# Validation of a Custom Interface Pressure Measurement System to Improve Fitting of Transtibial Prosthetic Check Sockets

**DOI:** 10.3390/s23073778

**Published:** 2023-04-06

**Authors:** Lucy Armitage, Kenny Cho, Emre Sariyildiz, Angela Buller, Stephen O’Brien, Lauren Kark

**Affiliations:** 1School of Mechanical, Materials, Mechatronic and Biomedical Engineering, University of Wollongong, Wollongong, NSW 2522, Australia; 2Graduate School of Biomedical Engineering, University of New South Wales, Sydney, NSW 2052, Australia; 3Orthopaedic Appliances, Pty, Ltd. (OAPL), Alexandria, NSW 2015, Australia; 4Tyree Foundation Institute of Health Engineering, University of New South, Sydney, NSW 2052, Australia

**Keywords:** prosthetic socket, interface pressure, amputation

## Abstract

Achievement of fit between the residual limb and prosthetic socket during socket manufacture is a priority for clinicians and is essential for safety. Clinicians have recognised the potential benefits of having a sensor system that can provide objective socket-limb interface pressure measurements during socket fitting, but the cost of existing systems makes current technology prohibitive. This study will report on the characterisation, validation and preliminary clinical implementation of a low cost, portable, wireless sensor system designed for use during socket manufacture. Characterisation and benchtop testing demonstrated acceptable accuracy, behaviour at variable temperature, and dynamic response for use in prosthetic socket applications. Our sensor system was validated with simultaneous measurement by a commercial sensor system in the sockets of three transtibial prosthesis users during a fitting session in the clinic. There were no statistically significant differences between the sensor system and the commercial sensor for a variety of functional activities. The sensor system was found to be valid in this clinical context. Future work should explore how pressure data relates to ratings of fit and comfort, and how objective pressure data might be used to assist in clinical decision making.

## 1. Introduction

Achievement of fit between the residual limb and prosthetic socket during socket manufacture is a priority for clinicians and prosthesis users, and is essential for safety and end-user satisfaction [1]. Poor fit can result in comfort or skin problems on the residual limb, as well as causing secondary problems elsewhere on the body, such as altered gait biomechanics [2,3]. One study has shown that of those fitted with a transtibial prosthesis, 48% identify socket fit as the main factor affecting rehabilitation. In the same study, 65.7% of clinicians also rated socket fit as the largest factor affecting rehabilitation [4].

There are many factors affecting socket-limb fit. Limb volume fluctuations with different activities, mechanical interaction between the limb and the socket, the temperature of the limb inside the socket, and prosthetic hardware all play a role [5,6]. Altered sensation in the residual limb may also mean that subjective feedback from the prosthetic user may not identify problems reliably [7].

Several sensors have been developed that can measure pressure at the socket-limb interface [8,9]; however, to date, these have been used to compare different prosthetics or activities, rather than to provide clinicians with information about fit during the socket design process. Furthermore, difficulties with cost, bulk, wires, and limitations in range or sensor performance have limited the translation of these systems into the clinical space for day-to-day use by prosthetists [10].

For transtibial residual limbs, different socket designs will load and offload selected regions of the residual limb anatomy. Specific surface bearing (SSB) sockets selectively load regions of the residual limb that are more pressure tolerant, for example the patella tendon in a patella tendon bearing (PTB) socket [11,12,13]. By selecting smaller regions to bear load, these localised regions are subjected to larger magnitudes of loading and can suffer from soft tissue injury [5]. Total surface bearing (TSB) sockets aim to distribute the load over the entire surface of the residual limb to reduce the magnitude of local stresses. [14,15,16]. In general, pressure tolerant areas of the transtibial residual limb include the patella tendon, the medial flare of the tibia, the residual pretibial musculature, the fibular shaft, the popliteal area, and the gastrocnemius muscle belly. Pressure intolerant areas include the tibial crest, tibial tuberosity, the distal ends of the tibia and fibula, the head of the fibula and peroneal nerve, and the hamstring tendon and patella [17].

We have previously reported a study where pressure sensor feedback was used to inform socket modifications [18]. This study showed that feedback to clinicians during check socket fitting was useful in reducing pressures over specific anatomical landmarks and improving between clinician consistency. In a recent survey of clinicians regarding challenges and management strategies during the socket manufacturing workflow, the check socket stage was identified as one where prosthetists experience challenges with socket fitting and have limited strategies available to assist the with these challenges [19]. In the same study, when asked what sort of technologies may assist with fitting at this stage of the workflow, prosthetists identified ways to accurately measure pressure between the socket and residual limb as a priority. This presents an opportunity for the development of a pressure sensing system that is designed to assist with socket fitting at the check socket stage of the workflow specifically. In check socket fit assessment, clinicians routinely use a ball of plasticine to assess the loading on the distal end of the residual limb. This is generally a less pressure tolerant region of the limb, and it is therefore important that pressure there is not too high. In a vacuum socket, it is also important that there is some contact between the residual limb and socket at the distal end to reduce the chance of a negative pressure air cavity that can draw additional fluid into the distal end of the limb and cause soft tissue injury such as verrucous hyperplasia.

Check sockets are typically made prior to the definitive socket being manufactured. They are made from a thermoplastic that is easy for the clinician to modify using a heat gun. Pressure measurement during the check socket stage offers the advantage of providing clinicians with pressure data at a time when they can make socket geometry modifications based on the pressure measurements. The clinical requirements for a sensor system for use at the check socket stage of socket fitting include a system that is safe, quick to install and remove, low in weight, has a wireless capability, and is simple to use in terms of operation and data interpretation. Based on the distal end of the residual limb being a common region of interest in vacuum check socket fitting, this region was selected for pressure sensing in this study. The chief contributions of this paper are to report the characterisation, validation and preliminary clinical implementation of a sensor system that has been designed to meet these user requirements for use during check socket fitting.

## 2. Materials and Methods

### 2.1. Sensor System Development

Our wireless, portable sensor system uses off-the-shelf single element capacitive sensors with a force range of 0–45 N and a circular sensing area with a diameter of 15 mm (equivalent pressure of 0–250 kPa) (SingleTact, PPS UK Limited, Glasgow, UK). The system is capable of simultaneously measuring normal pressure at up to four anatomical locations that can be determined by the clinician at the time the system is installed in the socket. Each module of the system is outlined in Figure 1.

#### 2.1.1. Portable Electronics Module

Each pressure sensor includes an interface circuit board that amplifies and converts to a I2C digital interface. The sensors were connected to an Arduino Nano (Arduino, Ivrea, Italy) and data were wirelessly transmitted via a pair of Wixel (Pololu, Las Vegas, NV, USA) radio modules to a serial monitor on a laptop PC. The system was powered with a mobile phone power bank. Pressure measurements from the four sensors were therefore transmitted across the clinical test area in real time over a distance of up to 15 m.

#### 2.1.2. Laptop Fixed Module

At the stationery end of the link, the receiving Wixel provided a standard serial connection into the PC. The PC can run a standard terminal program which continually displayed and tracked the measurements received, and the data were exported to a text file for storage and analysis.

#### 2.1.3. Firmware Description

The firmware was edited and compiled using the Arduino IDE running on Windows. Data from each sensor were printed with a timestamp. A baud rate of 57,600 (serial link speed) was selected to provide enough bandwidth for up to four sensors being sampled at 10 Hz.

### 2.2. Sensor Characterisation

Calibration was performed using four static loads throughout the sensor range of 0–45 N. Forces were applied via a benchtop rig and handheld weights, and sensor output was recorded.

As the system is to be used near the prosthetic liner and residual limb of the participants, the sensors were also assessed for alteration in a signal at higher operating temperatures by applying static loads, while sensors were positioned on a heat mat set to 40 °C, representing heating associated with body temperature. The rise and settling time and dynamic response to sinusoidal inputs at a variety of frequencies were also measured.

### 2.3. Benchtop Validation

Although our sensor system is capable of measuring the force at four locations simultaneously, for the purpose of this study, only one SingleTact sensor was used for benchtop validation and clinical testing. A commercial sensor, Loadpad (Novel, Munich, Germany) was used as the validation tool in this study. This sensor is a capacitive sensor with a force range of 0.1 N to 25 kN, an accuracy of 5 (% ZAS), and a sampling rate of up to 200 Hz. The sensing area is also circular with a 15 mm diameter, matching our sensor system. This system works via Bluetooth in conjunction with a proprietary mobile phone application. This application is used to calibrate the Loadpad sensors using an in-built process where the maximum expected load is entered into the software and then applied and removed to the sensor. The data are then transmitted wirelessly to the mobile application, where they are recorded.

For all subsequent sections of this study, the Loadpad was placed on top of our sensor, with simultaneous force measurements recorded on both sensors. Due to the differing material properties of the two sensors, it was necessary to explore their dynamic behaviour when on top of each other rather than in isolation. Compressive mechanical testing was performed using a Shimadzu UTM (EZ-X, Shimadzu). Two trials of the dynamic loading test were performed with the following loading condition: (1) SingleTact on top of Loadpad; and (2) Loadpad on top of SingleTact. Five cycles of 0–45 N were applied at two loading rates: 1 N/second and 5 N/second using a 500 N load cell. The hysteresis and average maximum force for each cycle was recorded.

### 2.4. Clinical Sensor Validation

#### 2.4.1. Participants

Three participants with transtibial amputation and a variety of years of prosthetic experience were recruited from private orthotic and prosthetic clinics. Participants were required to have had a transtibial amputation on only one of their legs for at least one year, and were deemed ready for a definitive socket fit by their prosthetists. Participants were also required to have their residual limb inspected by the prosthetist prior to participation to ensure that residual limb skin health was good and that there were no wounds, areas of redness, or skin damage. Finally, patients were required to have the ability to perform simple activities, such as standing up and walking, without the assistance of another person. Participants provided written and verbal informed consent for this study, which was approved by the Human Research Ethics and Clinical Trials at the University of New South Wales (HC200013).

#### 2.4.2. Experimental Design

A custom-built check socket was fabricated by a qualified prosthetist for each participant. This check socket was attached to the participant’s existing pylon and foot by the prosthetist and aligned according to the prosthetist’s usual methods.

The SingleTact and Loadpad were calibrated prior to collecting the data. Sensors were then inserted onto the inner wall of the distal end of the prosthetic socket for each participant. Sensors were taped in position at the distal end of the prosthetic socket. The SingleTact tail exited from the socket via a small drill hole in the side of the socket, and the Loadpad was attached to the inner wall of the socket such that the tail exited the socket on the posterior surface of the socket. Participants were then asked to don their prosthesis and then perform three common functional exercises that form a routine part of a socket fitting as far as they were able (Figure 2). During these activities, pressure data were simultaneously collected from the SingleTact and Loadpad. The activities were: (1) three repetitions of a 10 m walk; (2) three repetitions of the ascent and descent of three stairs; and (3) three sit-to-stands from a chair. All activities were performed in the clinical assessment room.

##### Ten Metre Walk

Participants were shown to a pre-marked 10 m track in the clinic. They were instructed to walk at a comfortable pace along the length of the track. This was repeated three times. The stance phase (initial contact to toe-off of the prosthetic foot) was extracted and presented as 0–100% of one cycle. A total of four cycles from each trial were extracted for analysis.

##### Stairs

Participants were asked to walk up and down three standard steps located in the clinical assessment room. Participants were instructed to use the handrail as required and to complete the activity at a comfortable pace. Three repetitions of climbing up and down the stairs were performed. The stance phase (initial contact to toe-off of the prosthetic foot) was extracted and presented as 0–100% of one cycle. A total of three cycles were extracted for analysis from each trial.

##### Sit to Stand

Participants were seated in a standard chair with arms and asked to stand from a seated position, hold a standing position for three seconds, and return to sitting. Participants were instructed to use their hands as required. No further prompting on foot position or weight distribution was given. One cycle was taken from the initiation of standing from the seated position to the return to sitting from the standing position. Three cycles were analysed for each participant.

### 2.5. Patient and Clinician Feedback

Upon the completion of all tasks, participants were asked to provide a rating of overall socket comfort, distal end comfort, and socket fit, as well as any feedback on the use of the sensor system during their appointment via a custom questionnaire containing Likert and free form questions. The clinicians were blinded to pressure measurements during the assessment. They were asked to rate the fit of the socket based on their usual clinical judgement.

### 2.6. Data Processing and Statistical Analysis

Data were processed using Matlab2015b (Mathworks, Natick, MA, USA). For each task, data were segmented and normalised as a percentage of the total task cycle to facilitate comparison within and between participants and sensors. Mean and SD across all trials were calculated for each sensor during each task. Statistical Parametric Mapping (SPM) was used to perform a paired t-test and to assess the statistical significance of differences in the magnitudes of the measured forces between the Loadpad and the SingleTact [20]. SPM processes produce a figure with a threshold value of the t statistic when alpha = 0.05, represented by the red dotted line in the SPM plot. If the t value crosses this threshold at any stage on the figure, a statistically significant difference is deemed to exist [20].

The differences between the mean force value for the SingleTact and Loadpad throughout the cycle were calculated as an absolute value.

## 3. Results

### 3.1. Sensor Characterization

Static testing using a benchtop rig and hand weights was performed at room temperature and 40 °C (Figure 3).

A least squares regression line was used to generate a calibration function at room temperature with an R^2^ value of >0.99 for each sensor. The peak difference between sensor output at room temperature and 40 °C was 1.57 N, corresponding to a 3.5% error with respect to the full sensor range. Table 1 shows the percentage of drift for the three SingleTact sensors with constant loading for 1 min at each mass. Peak drift was 1.45% in the middle of the range for sensor 2.

The dynamic response of the force sensor is illustrated using step and sinusoidal force inputs in Figure 3 and Figure 4, respectively. As shown in Figure 4b, the rise time of the sensors is less than 40 ms, while its settling time is less than 65 ms. The sinusoidal force input response illustrated in Figure 5 shows that the bandwidth of the force sensor is over 10 Hz. We observed in experiments that the accuracy of force measurement significantly deteriorated after 6 Hz.

### 3.2. Benchtop Validation

Figure 6 shows the hysteresis behaviour for the SingleTact at two different loading rates and stacking conditions. Hysteresis errors were calculated at the midpoint of the applied force range (22.5 N) as a percentage of the difference between ascending and descending output divided by the overall tested range. These were 1.3% when the SingleTact was placed on top of the Loadpad at 1 N/s, 0.93% when the Loadpad was placed on top at 1 N/s, 3.51% when the SingleTact was placed on top at 5 N/s, and 5.3% when the Loadpad was placed on top at 5 N/s. Loadpad hysteresis also varied between loading conditions and stacking order, with a range of 0.7–7.5% across all tests. The behaviour of the sensor over the sensing range as well as the maximum recorded values did not show differences for different loading rates or stacking conditions.

### 3.3. Clinical Implementation and Validation

Three participants consented to participate in the study, and the participant demographics are summarised in Table 2. The sensor system was installed in the distal end of three sockets of differing shapes and sizes (Figure 7A) in conjunction with the Loadpad sensors. The full system installed on a socket can be visualised in Figure 7B. Sensors were placed in the prosthetic socket to measure the force at the interface between the distal end of the residual limb and the prosthetic socket for each participant, as shown in Figure 7C. All participants had total surface bearing sockets with a passive vacuum suspension and knee sleeve. Participants 1 and 3 had a silicone cushion liner, while participant 2 had a polyurethane liner. Participant 1 had an Aeris Performance foot, participant 2 had an Ottobock Trias foot, and participant 3 had an AllPro foot.

During the 10 metre walk, mean peak force from the SingleTact and Loadpad were 22.89 N and 23.43 N, respectively, for participant one, 19.97 N and 19.56 N, respectively, for participant two, and 21.01 N and 21.79 N, respectively, for participant three. Despite the differences in the measured peak mean forces found between two sensors, the SPM analysis showed that there were no statistically significant differences between the mean force values of the two sensors throughout the ten metre walks (Figure 8), because the t statistic value (black line on SPM plot) never crosses the threshold value for the t statistic when alpha = 0.05 (red dotted line on SPM plot).

The differences in measured force between the SingleTact sensor and the Loadpad sensor fluctuated throughout the cycle. The magnitude of the differences between sensors varied from 0 N to 6 N between participants.

Sit-to-Stand exercise presented a broader range of measured mean forces from two sensors between participants, ranging from 3 N to 13 N. For sit to stands, the mean peak force from the SingleTact and Loadpad were 12.46 N and 12.16 N, respectively, for participant 1, 2.87 N and 2.9 N, respectively, for participant 2, and 12.25 N and 11.76 N, respectively, for participant three. There was no clear trend in measured mean force values and the type of sensor, as the measured forces fluctuated between two sensors throughout the cycle for all participants. However, the SPM analysis showed that there was no statistically significant difference between the measured mean force values between two sensors for all participants. Despite no statistically significant differences between the two mean forces, the quantitative differences in mean values ranged from 0 N to 5 N between participants. All the measured mean force curves presented concave-down shapes, where the peak mean forces were found at 50% of the total cycle.

Figure 9 shows force measurements for stair ascent (left) and descent (right) for two participants (the third was unable to climb stairs at the time of the appointment). For stair ascent, the mean peak force from the SingleTact and Loadpad were 17.89 and 21.5 N, respectively, for participant one, and 19.22 N and 22.03 N, respectively, for participant three. Despite the differences found in the peak mean force values, the SPM analysis showed that there were no significant differences in the mean forces of two sensors for both participants. However, the differences in mean forces fluctuated for each participant. Participant one showed a greater range of mean force differences, ranging from 0 N to 5.6 N, while the differences of the mean force of participant three ranged from 0 N to 3.2 N. There was no clear pattern in these differences in mean forces over the total cycle.

For stair descent, the mean peak force from the SingleTact and Loadpad were 18.33 N and 21.89 N, respectively, for participant one, and 18.11 N and 19.48 N, respectively, for participant three. The SPM analysis showed that there were no significant differences found between the two measured mean forces of two sensors throughout the cycle. The quantitative differences between these mean forces varied between two participants. Participant one showed a greater range of difference in measured mean forces, ranging from 0 N to 4.7 N, while the differences of the mean force of participant three ranged from 0 N to 3.3 N.

### 3.4. Participant and Clinician Feedback

All prosthesis users gave a score from 0 to 10 to describe overall socket comfort, distal end comfort, and socket fit. All participants gave a maximum rating of 10 in all categories (Table 3). The Qualitative feedback from participants indicated that, universally, all participants found the socket very comfortable to use during all exercises. Most of the participants noted that our sensor system did not interfere with or restrict their movements during these exercises. However, prosthesis users suggested that further improvements regarding the usability of the device should be considered for future prototypes. Based on their usual assessment methods, all clinicians also rated the socket fit to be a good one. The table below shows the ratings and responses from participants in bullet points.

## 4. Discussion

This study has assessed the performance of a low cost, wireless in-socket pressure measurement device designed to measure distal end pressures in transtibial prosthetic check sockets. This device was designed as a simple additional objective measurement tool for prosthetists to use during check socket fitting in the clinic, something that has been called for in the professional community [19]. This sensor performed well during testing under the expected operating conditions, including temperatures up to approximately 40 °C, and dynamic loading during walking (the usual walking frequency for human walking is between 1.8 and 2 Hz [21]). Our system displayed comparable accuracy to the commercial system, with 3.5% errors between room and body temperature and hysteresis errors of approximately 5% (Loadpad, 5% [22]). Overall, the benchtop performance of SingleTact sensors was appropriate for use in the check socket application.

The peak SingleTact drift was within the expected performance range. However, the peak hysteresis error of 5.3% was higher than the expected range of <4% [23]. This error may be related to the stacking order the sensors, where the higher hysteresis error was observed when the Loadpad sensor was on top of the SingleTact sensor. This stacking order was necessary in our application, as the adhesive tapes used to stabilise the sensors caused damage to the sensing surface of the SingleTact when it was placed on top. This is a limitation in this study, and future studies using the SingleTact in isolation should consider the adhesion method to ensure that the sensor integrity is maintained without affecting the dynamic behaviour. Furthermore, the curved surface and soft interface with limb tissue may also have resulted in less robust force readings than on a rigid surface [24].

In clinical testing, we found no statistically significant differences in sensor output between the SingleTact and Loadpad sensors across the sit to stand, walking, and stair ascent and descent exercises for three participants with different socket geometries and body weights. Forces measured by the SingleTact and Loadpad varied at some points in the cycle by up to approximately 5 N. If this reflects a difference in the force measurement of each system, this magnitude of difference may be clinically significant. It is important to note, however, that when considering the peak values that each sensor measured throughout the cycle, the differences were of a much smaller magnitude (in the range of 1 N). Therefore, it is likely that differences in values between sensors are due to differences in dynamic response and data synchronisation rather than reflecting a true difference in the magnitude of the force measured. In clinical practice, the peak values will likely be the parameter of interest, and the high level of agreement between these for the Loadpad and SingleTact would suggest that they can be used interchangeably in this context.

The clinical force magnitudes do not provide any insight into socket fit in isolation, and must be considered in the context of ‘good fit’ as reported by the clinicians and the prosthetic user. Thresholds for ‘good fit’ with regard to force or pressure have not yet been established in the literature [25]. In this study, all participants reported full satisfaction with their sockets at the time of fitting, in agreement with the clinician assessments. Our force measurements did not exceed 24 N at the distal end of the socket, perhaps suggesting that this magnitude is within the ‘comfortable’ range. However, this force measurement corresponds to a pressure of 135 kPa, which is higher than the threshold above which tissue damage may be occurring [26]. Given the small sample size, more robust studies should investigate this further, linking force or pressure measurements to comfort scales as well as to skin health.

Our sensor system has several limitations. Firstly, the implementation of the SingleTact system in the check sockets requires a small drill hole to be made into the socket adjacent to the point(s) where measurement will be made. This has the potential to alter the loading state close to the hole, and also means that there is permanent damage done to the socket. In the context of a check socket that is a temporary measure, this modification may be acceptable in order to take the pressure measurements, but this may not be the case in definitive sockets. End users also provided feedback about the size of the device, stating that it would need to be smaller and less intrusive if it was to be worn for longer-term measurements. Secondly, the SingleTact data is currently logged as a text file and post-processed. This process is time consuming and difficult to implement in the clinic; therefore, a mobile application or computer program to record and visualise data is required before this system can be implemented in the clinic. Thirdly, the small sample size and limited data collected mean that these results cannot be generalised, and more work needs to be done to better clarify the relationship between objective force or pressure measurements and socket fit and comfort. Future works should also examine the use of pressure data to inform clinical decision making and the effect of socket modifications on pressure measurements at the socket-limb interface. Finally, all participants used a total surface bearing socket with a passive vacuum and knee sleeve; however, other prosthetic components varied. Variations in prosthetic components may affect the interface pressures during data collection. In future studies where the collected data is informing the clinical interpretation of fit, componentry should be considered in interpretation, and normative interface force or pressure values for different prosthetic set ups may be required. The activity level of participants may also affect which prosthetic components they are fitted with, and therefore interface pressures as well. This should also be considered in future works when examining interface pressures between the socket and residual limb.

## 5. Conclusions

In this study, a portable, wireless force measurement system was characterised and validated at the socket limb interface of three participants with transtibial amputation. Sensor characterization determined that it was suitable for this application. Clinical validation found no statistically significant differences between it and a commercial sensor system during walking, stairs, or sit to stand activities. Future works should explore how pressure data relates to ratings of fit and comfort, and how objective pressure data might be used to assist in clinical decision making.

## Figures and Tables

**Figure 1 sensors-23-03778-f001:**
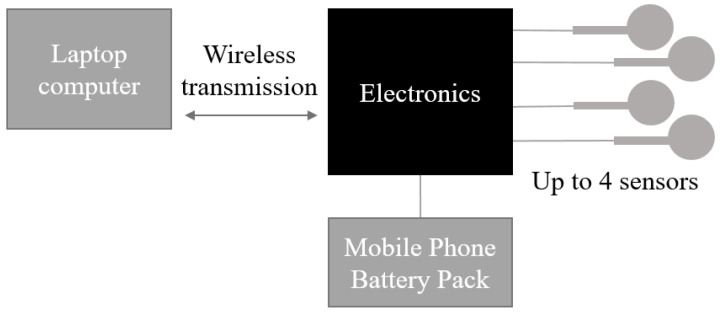
Schematic of an in-house wireless portable sensor system. The electronics module contains the Arduino Nano and transmitting Wixel and is powered by a standard USB power bank. The four sensors are connected via four I2C bus connectors.

**Figure 2 sensors-23-03778-f002:**
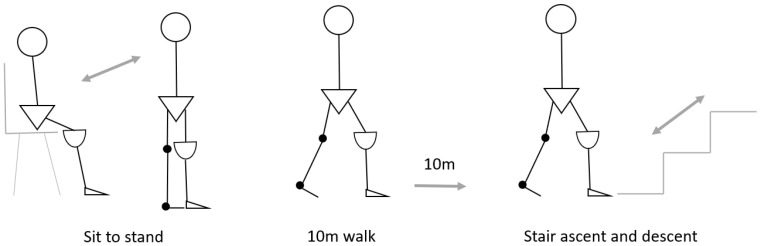
Description of three functional activities where pressure measurement were performed.

**Figure 3 sensors-23-03778-f003:**
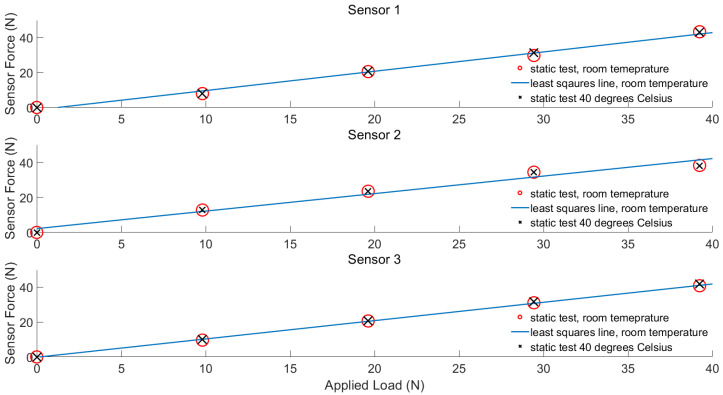
Static testing at room and body (40 °C) temperatures for three different SingleTact sensors.

**Figure 4 sensors-23-03778-f004:**
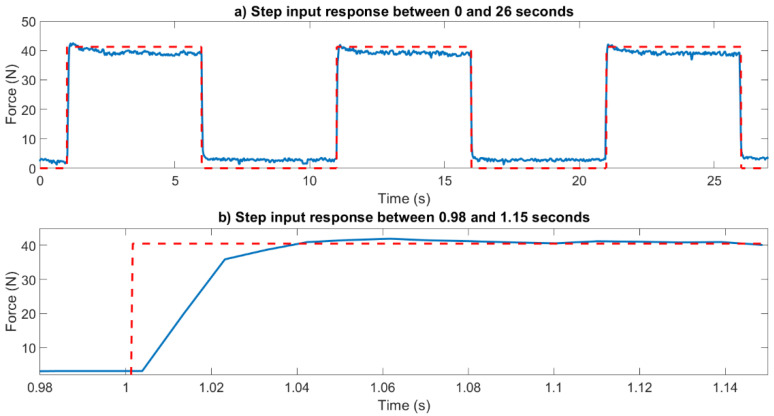
(**a**) Step input response between 0 and 26 s, (**b**) step input response between 0.98 and 1.15 s. The red line shows the input and the blue line shows the sensor response.

**Figure 5 sensors-23-03778-f005:**
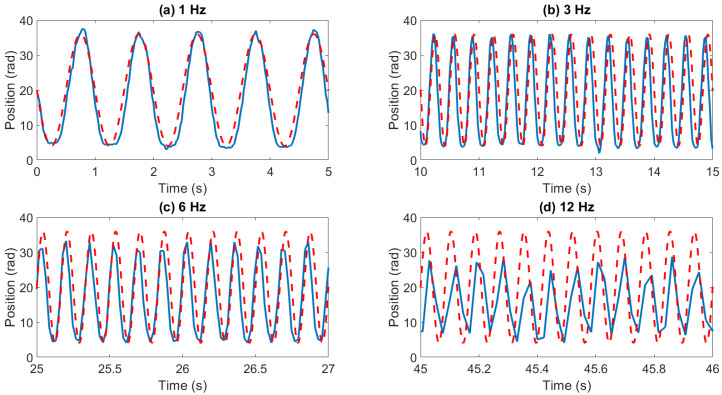
Sensor response (blue line) to sinusoidal input (red line) at (**a**) 1 Hz, (**b**) 3 Hz, (**c**) 6 Hz and (**d**) 12 Hz. The red line shows the input and the blue line shows the sensor response.

**Figure 6 sensors-23-03778-f006:**
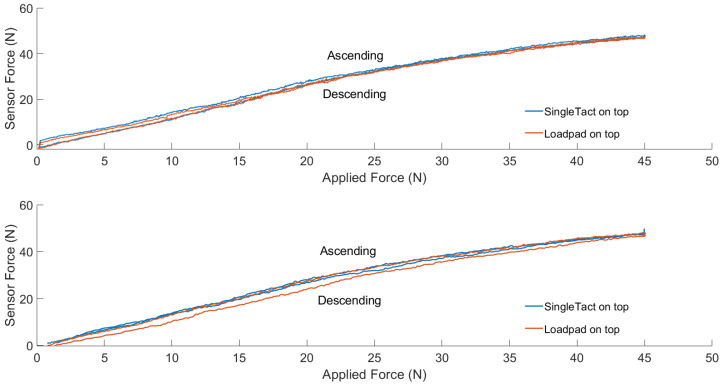
Hysteresis of the SingleTact and the Loadpad sensors at two different loading rates and stacking order.

**Figure 7 sensors-23-03778-f007:**
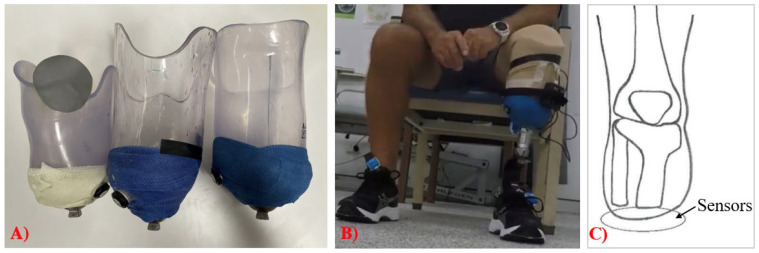
Experimental setup: (**A**) participants’ check sockets, (**B**) participant wearing the sensor system, (**C**) sensor placement on the distal end of a residual limb, where one SingleTact and one Loadpad force sensor were placed.

**Figure 8 sensors-23-03778-f008:**
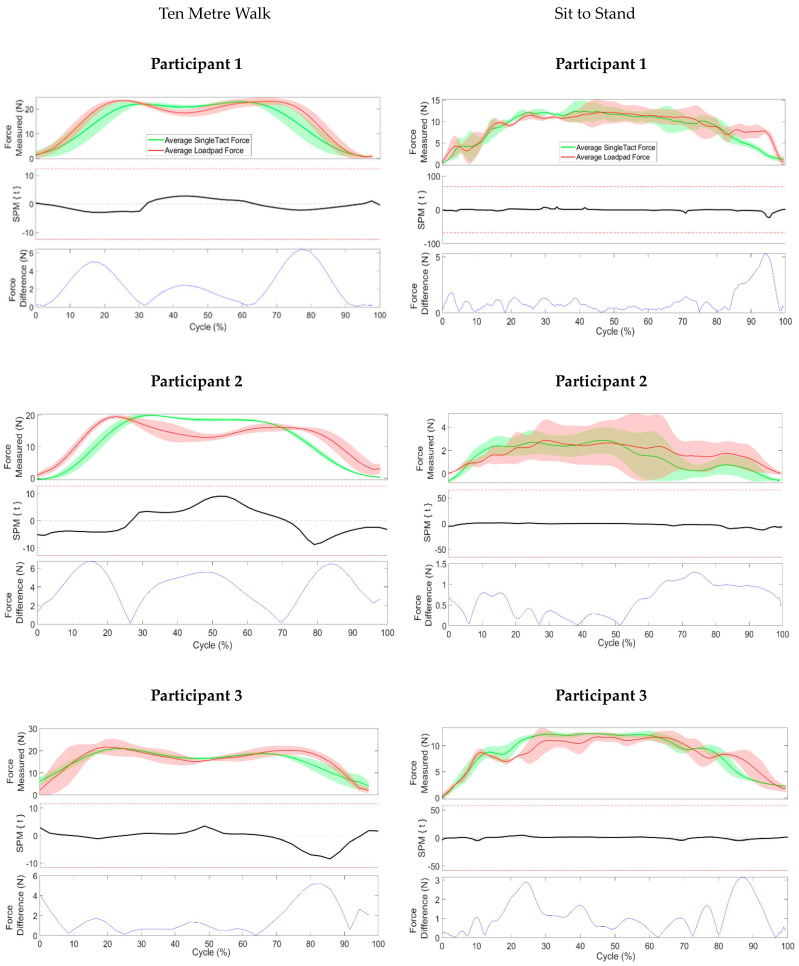
Force measurements from SingleTact and Loadpad at the distal end of the residual limb during walking (**left** column) and sit to stand (**right** column). The top graph for each participant represents the force measurements from the SingleTact (green) and Loadpad (red), the middle graph represents SPM (Mean and SD) of the difference between sensor measurements, and the bottom graph represents the magnitude of the difference between sensor measurements throughout the cycle.

**Figure 9 sensors-23-03778-f009:**
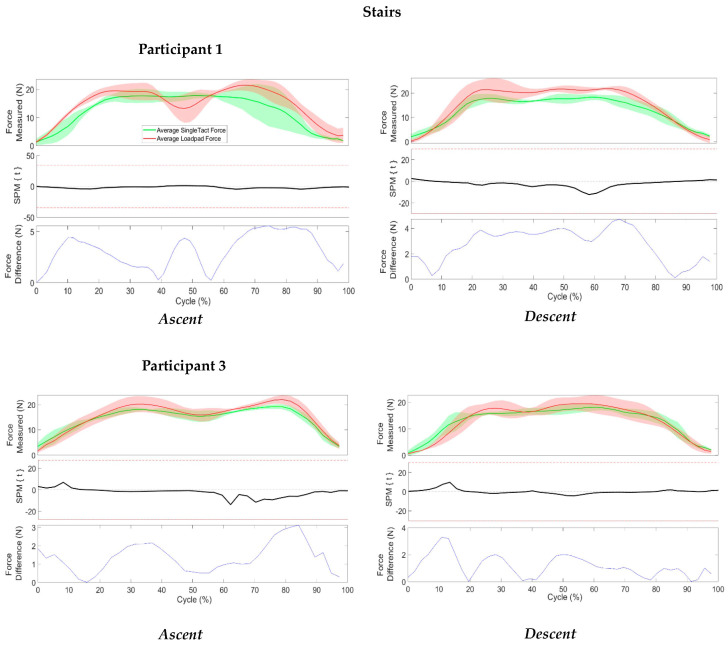
SPM (Mean and SD) of force measurements for SingleTact and Loadpad at the distal end of the residual limb during the ascent and descent of stairs: (green = measured average force of SingleTact, red = measured average force of Loadpad).

**Table 1 sensors-23-03778-t001:** Percentage drift over 1 min constant loading.

Weight (N)	Sensor 1 (%)	Sensor 2 (%)	Sensor 3 (%)
9.81	0.8	0.6	0.36
19.62 N	0.88	0.7	1.45
29.43 N	0.3	0.53	0.75
39.24 N	0.4	0.07	0.01

**Table 2 sensors-23-03778-t002:** Summary of participant demographics.

Participant	Age (Years)	Gender	Side and Type of Amputation	Weight (kg)
1	64	Male	Right transtibial	83
2	75	Male	Left transtibial	115
3	72	Male	Left transtibial	90

**Table 3 sensors-23-03778-t003:** Participants’ feedback with ratings for each criteria, ranging from 0 (not at all comfortable) to 10 (extremely comfortable).

	P1	P2	P3
Overall Comfort Rating (0–10)	10	10	10
Distal End Comfort (0–10)	10	10	10
Socket Fit Rating (0–10)	10	10	10
Qualitative Feedback	“The socket felt comfortable and was not able to feel any pressure at the distal end” “The hardware of the sensors could be improved”	“The socket fit perfectly without any pain”	“The socket was comfortable to wear, and it fit firmly.”

## Data Availability

The data presented in this study are available on request from the corresponding author. The data are not publicly available due to data sharing restrictions in ethics approval.

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
