# Peer review of "Validation of a Custom Interface Pressure Measurement System to Improve Fitting of Transtibial Prosthetic Check Sockets"

_sensors, 2023, doi:10.3390/s23073778_

Round 1

Reviewer 1 Report

The authors present an article dealing with validation of a custom interface pressure measurement system to improve fitting of transtibial prosthetic check sockets.

Introduction

In the introduction, I lack information about loadable and non-loadable places on the residual limb, which places are more prone to damage. Mention the studies that were concerned with the measurement of pressure in these places. 

Reasoning, why you chose the distal part of the stump, when it should be the least stressed place when using the prosthesis.

Materials and Methods

How did you ensure that the sensor does not move during the measurement?

When measuring the pressure, were all three types of prostheses the same? Did they have the same type of socket (if yes, what type)? Did they have the same knee and ankle joint mechanisms? 

What was the physical activity of the patients? In the mentioned study, the patients are older. What was the condition of the stump - wounds, redness, skin damage.....?

Perhaps a picture of the individual activities during the pressure measurement would be appropriate.

Result

Units are written in square brackets (figures and tables throughout the chapter). 

Table 2 - I might add how long patients use the prosthesis

Reviewer 2 Report

This paper aims to develop a built-in pressure sensor at the prosthetic/body interface and compares it to a commercial sensor. Overall, the manuscript is well-written and requires minor clarifications.

1) In general, a statistical analysis section would help to understand the data a little better, especially on SPI analysis and its interpretation of significance.

In Fig.7, for example, there seems to be a difference in the forces measured between the commercial and custom sensors for participant #2 ; however, the authors claim that this is not statistically significant based SPI analysis, which is hard to follow.

2) Minor spellcheck is required (i.e Fig.2: 'Celcius?)

3) Ln233-235 seems irrelevant.

4) Was the same type check socket used for all participants? There are many types of transtibial prosthetic sockets (i.e Patella Tendon Bearing, PTB SC, SocketSSS etc)). Which one was used here? How does the type of prosthetic impact sensor readings?
